# Is There a Connection between the Metabolism of Copper, Sulfur, and Molybdenum in Alzheimer’s Disease? New Insights on Disease Etiology

**DOI:** 10.3390/ijms23147935

**Published:** 2022-07-19

**Authors:** Fábio Cunha Coelho, Giselle Cerchiaro, Sheila Espírito Santo Araújo, João Paulo Lima Daher, Silvia Almeida Cardoso, Gustavo Fialho Coelho, Arthur Giraldi Guimarães

**Affiliations:** 1Laboratório de Fitotecnia (LFIT), Universidade Estadual do Norte Fluminense Darcy Ribeiro—UENF, Campos dos Goytacazes 28013-602, Brazil; 2Centro de Ciências Naturais e Humanas, Universidade Federal do ABC, Av. dos Estados, 5001, Bl. B, Santo André 09210-170, Brazil; giselle.cerchiaro@ufabc.edu.br; 3Laboratório de Biologia Celular e Tecidual (LBCT), Universidade Estadual do Norte Fluminense Darcy Ribeiro—UENF, Campos dos Goytacazes 28013-602, Brazil; espiritosanto.sheila@uenf.br (S.E.S.A.); agiraldi@uenf.br (A.G.G.); 4Departamento de Patologia, Hospital Universitário Antônio Pedro, Universidade Federal Fluminense, Niterói 24210-350, Brazil; jpldaher@id.uff.br; 5Departamento de Medicina e Enfermagem (DEM), Universidade Federal de Viçosa, Viçosa 36579-900, Brazil; silvia.cardoso@ufv.br; 6Instituto de Ciências Médicas, Universidade Federal do Rio de Janeiro, Macaé 27930-560, Brazil; gustavofialhoc@gmail.com

**Keywords:** Alzheimer’s disease, metal imbalance, molybdenum, copper, sulfur

## Abstract

Alzheimer’s disease (AD) and other forms of dementia was ranked 3rd in both the Americas and Europe in 2019 in a World Health Organization (WHO) publication listing the leading causes of death and disability worldwide. Copper (Cu) imbalance has been reported in AD and increasing evidence suggests metal imbalance, including molybdenum (Mo), as a potential link with AD occurrence.We conducted an extensive literature review of the last 60 years of research on AD and its relationship with Cu, sulfur (S), and Mo at out of range levels.Weanalyzed the interactions among metallic elements’ metabolisms;Cu and Mo are biological antagonists, Mo is a sulfite oxidase and xanthine oxidase co-factor, and their low activities impair S metabolism and reduce uric acid, respectively. We found significant evidence in the literature of a new potential mechanism linking Cu imbalance to Mo and S abnormalities in AD etiology: under certain circumstances, the accumulation of Cu not bound to ceruloplasmin might affect the transport of Mo outside the blood vessels, causing a mild Mo deficiency that might lowerthe activity of Mo and S enzymes essential for neuronal activity. The current review provides an updated discussion of the plausible mechanisms combining Cu, S, and Mo alterations in AD.

## 1. Introduction

Alzheimer’s disease (AD) is the most common form of dementia globally in older adults [1]. It is highly inheritable, and diet and lifestyle are influential for increased risk [2]. Histopathologically, AD is characterized by amyloid β plaques (Aβ) and tau neurofibrillary tangles in the brain [3]. The world scientific community has been studying AD for decades, with some research lines investigating copper (Cu) and sulfur (S) metabolisms and, more recently, molybdenum (Mo) metabolisms.

Cu is an essential element for the functioning of enzymes linked to oxidative stress scavenging. The alteration of its homeostasis is associated with free radical production and cellular damage [4]. In the body, S is found in mineral form, sulfate (SO_4_^2^^−^) being also present in organic molecules such as vitamins, coenzymes, and S-amino acids [5]. Mo-containing enzymes are found in human and animal tissues, and the redox potential of MoV/MoVI is suitable in redox chemistry involving electron exchange with flavin mononucleotides [6].

Many years ago, there were reports that the altered metabolism of S amino acids occurs in AD [7]. The relationship between Cu imbalance and AD is already well reported in the scientific literature [8]. Furthermore, there are many similarities between some metabolic consequences of Mo deficiency [9,10] and AD.

## 2. Linking AD Predisposing Factors to Cu and S Metabolism

Pathologies or symptoms related to AD usually occur before and/or concomitantly with the onset of symptoms, such as memory loss. Multiple factors are associated with AD, including hypertension, diabetes, hypercholesterolemia, obesity, and depression [1], as well as sex hormones [11] and hyperhomocysteinemia [12], all of them linked in some way to Cu and S disturbances.

Consistently, people with low performance in the Digit Symbol Substitution cognitive test tended to have a higher prevalence of hypertension and diabetes, among other diseases, than people with average cognitive performance. It is of particular interest that cognitive performance in the same test decreased with an increasing total Cu intake, indicating that AD-linked risk factors could be correlated to altered Cu levels [13].

Cu can influence blood pressure regulation via tyrosinase activity, a key enzyme in synthesizing norepinephrine, a major vasoconstrictor factor [14]. Therefore, hypertension can be affected by changes in Cu concentration [14]. Additionally, a multinomial logistic model revealed that an increase of non-Ceruloplasmin Cu (nCp–Cu)in one-unit standard deviation raised the relative risk of having Type 2 Diabetes (T2D) by 9.64 [15]. Moreover, serum levels of Cu^2+^ are significantly higher in rat models of depression [16] and significant depression patients [17] compared to healthy controls [18]. Interestingly, Cu exposure exacerbates the depression-like behavior of ApoE4 mice. The mechanisms may involve the dysregulation of synaptic function and immune response and the over-activation of neuroinflammation [19].

High Cu levels have been shown influence negatively testosterone synthesis and body levels [20,21]. Testosterone levels are significantly lower in AD brains than in control ones [22]. Brain testosterone levels are also considerably reduced in men with mild neurological changes (MNC) consistent with early-stage AD. Rosario et al. [22] evaluated brain tissue at autopsy of three groups: control subjects, subjects with MNC, and subjects with AD, all with similar ages (mean (SD) ages of 70.9 (5.3), 72.9 (5.7), and 72.0 (6.5) years old, respectively). They found 1.0, 0.6 and 0.5 ng g^−1^ of wet tissue of testosterone, respectively. In comparison with the testosterone curve, urinary Mo and serum Cu were inversely associated in a data survey, including men aged 18–55 years from NHANES 2011–2012 [23].

Changes in S compound metabolism seem implicate on the AD risk factors depression and hyperhomocysteinemia. An unhealthy dietary pattern has been associated with an increased risk of depression via decreased serum folate contents [24]. Folate is a cofactor required for S-containing amino acid metabolism, particularly important forproducing methionine (Met) from Homocysteine (Hcy), avoiding hyperhomocysteinemia. The high levels of Hcy are involved in the decreased synthesis of catecholamines and depression [24] and predisposition to AD [25]. Namekawa et al. [26] evaluated depression in the elderly and found an increased risk of AD and Aβ protein metabolism.

The accumulated knowledge in the last decades has provided significant advances in understanding AD. Furthermore, the relationship of AD with Cu and S metabolism and other AD-associated diseases has had much advancement but still does not allow for a clear explanation of AD’s causes. Squitti et al. [8], in a recent review, commented in-depth about the Cu imbalance in AD and its link with the Aβ hypothesis. However, they did not mention S metabolism.

## 3. Linking Cu to AD Pathology or Multiple AD-Associated Factors

Total body Cu in adult humans has been estimated to range from 70 to 150 mg, with the highest contents in the liver, brain, heart, and kidneys [27]. Cu is a nutrient for humans, which participates in the composition of essential proteins: ceruloplasmin, Cu–Zn superoxide dismutase, nitric reductase, amine oxidase, tyrosinase, bilirubin oxidase, Cytochrome c oxidase (COX), and plastocyanin, for example [28]. The main route of excretion of Cu from the body is bile [29]. However, it can lead to oxidative stress and other harmful effects when in excess.Healthy total Cu serum ranges are 700 µg L^−1^–1300 µg L^−1^ [30].

The daily nutritional recommendations and maximum tolerable levels of Cu intake for people above 19 years of age are the same regardless of age [27,31]. More studies should be conducted to determine the Cu amount needed and the maximum limits by age groups over 19 years old. It is possible that at older ages Cu intake limits should be lower than inyounger generations because Cu can accumulate in the body with aging [32,33,34]. However, more than the Cu accumulation, it is the dislocation of loosely bound Cu in the brain that can cause toxicity [35,36]. In addition, some foods may increase Cu content due to the use of Cu to control plant diseases in agriculture [37].

Masaldana et al. [38] commented that cell senescence is characterized by irreversible growth arrest caused by replicative exhaustion or pro-oncogenic cell stressors (radioactivity, oxidative stress, oncogenic activation). Additionally, the same authors reported that enrichment with metals, including Cu, in the senescent cells from tissues during aging had been associated with tissue dyshomeostasis and age-related pathologies, including cancer and neurodegenerative disorders (AD, Parkinson’s disease (PD)), and metabolic disorders (e.g., diabetes). Interestingly, Masaldana et al. [38] identified Cu accumulation as a universal feature of senescent cells (fibroblasts from embryonic mice, human prostate epithelial cells, and human diploid fibroblasts) in vitro. In fact, in vitro experiments showed the exposure of fibroblasts to Cu sulfate, which increases intracellular Cu, induces premature senescence through oxidative stress [39]. This fact indicates a specific association between the accumulation of Cu and the anomalies mentioned. On the other hand, a major free-radical chain-breaking antioxidant, vitamin E (α-tocopherol), may interfere in the initiation and progression of Cu-induced oxidative damage [40].

Chronic exposure to Cu and its dyshomeostasis has been linked to accelerated cognitive decline and potentially increased risk of AD [41,42,43]. Furthermore, Cu ions, due to their redox ability, have been considered to be potential therapeutic targets in AD, and a considerable number of ligands have been developed to modulate the toxicity associated with Cu in this context via disruption of the Aβ–Cu interaction [44]. Heng-Wei et al. [45] reported that recent evidence suggests that dyshomeostasis of Cu and its valency in the body are major determinants of its beneficial effects as an essential metal or its neurotoxic counterpart. Following this line, James et al. [46] appraised redox-competent Cu species (Cu^2+^ and Cu^1+^) levels in AD. They quantified the burden of redox-active labile Cu in postmortem brain tissue of AD subjects and age-matched cognitively normal controls. This study found high contents of exchangeable Cu^2+^ and increased capacity of AD cortical tissue samples to bind Cu^2+^.

In healthy people, 90–95% of the Cu circulation is bound to the Cu-binding protein ceruloplasmin [47]. NCp–Cu characterizes AD patients, and possible genetic defects of the ATP7B gene, which encodes for a P-type ATPase protein involved in the Cu transport, were identified in AD patients [48]. In AD, Cu decreases in the brain but increases in the serum due to the nCp–Cu component increase [36,49]. Excess Cu speeds up some early AD events, such as the induction of aggregation of Aβ peptides via enhancing the amyloidogenic cleavage of amyloid precursor protein (APP) and the hyperphosphorylation of Tau by stimulating GSK3β kinase, disturbing brain function of wild-type mice and exacerbating neurodegenerative changes in a mouse model of AD [50]. Therefore, studies have recently tested chelators of Cu as therapeutic approaches in the AD animal models or have even used Cu to induce AD animal models [51].

## 4. Linking S Metabolism to AD Pathology or Multiple AD-Associated Factors

In humans, S inorganic species, such as SO_4_^2−^, and S-biomolecules, including S-containing amino acids, their derivatives, S-vitamins, biotin or S-coenzymes, and co-factors, as coenzyme A and α-lipoid acid, are obtained from a regular and equilibrated diet [5]. Met is an essential S amino acid that functions as a precursor for the synthesis of Hcy and cysteine (Cys).

Varadarajan et al. [52] commented that Aβ is a source of oxidative stress from free radicals that can lead to neurodegeneration in AD. It may be partly due to the oxidative stress associated with free radicals from Aβ, which involves Met. Met-SO (methionine- sulfoxide), an oxidized form of Met, is an indicator of systemic oxidative stress. Weng et al. [53] found serum levels of Met-SO, significantly higher in early-stage AD patients than in mild cognitive impairment (MCI) patients.

In AD, there is in plasma an increase in the Cys content [7] and a decrease in SO_4_^2−^ [7], folate [54], and choline contents [55]. Moss and Waring [56] evaluated that plasma Cys/SO_4_^2−^ ratio is a possible clinical biomarker for investigating chronic health disorders including AD, PD, multiple sclerosis, myalgic encephalomyelitis, irritable bowel syndrome, autism, depression, motor neuron disease, and hemolytic anemia. It is interesting to note that although Cys contents are high in the plasma in AD [7], there is no consensus yet about its level inside the brain cells. While Luo et al. [57] found decreased Cys by ca. 3-fold in an animal model of AD, Mandal et al. [58] found higher ones using another detection method.

Individualswith low folate contents and high Hcy contents are at increased risk of AD [54,59,60,61,62]. Increased Hcy accompanies folate deprivation because folate is a necessary co-factor for the enzyme 5,10-methylenetetrahydrofolate reductase that mediates the conversion of Hcy to Met [63].

Irizarry et al. [64] investigated a link between high Hcy and AD. They found that elevated plasma total Hcy was positively correlated with plasma Aβ 40 and Aβ 42 levels. Additionally, prolonged hyperhomocysteinemia can lead to the onset of cardiovascular disease, atherosclerosis, stroke, inflammatory syndromes like osteoporosis and rheumatism, and neuronal pathologies, including AD and PD [65].

The relationship between folate and choline deficiencies with AD has been extensively studied [66,67,68,69]. Choline deficiency, for example, besides causing lower acetylcholine synthesis, contributes to mitochondrial dysfunction and, therefore, to the rate of fatty acid oxidation deficiencies and has also been linked to increased fatty acid uptake [69].

## 5. Hypothetical Relationship between Mo and AD Pathology or Multiple AD-Associated Factors

Mo is considered essential for animals and plants. It is a co-factor of enzymes (molybdoenzymes) xanthine oxidase-dehydrogenase (XO), aldehyde oxidase (AO), sulfite oxidase (SUOX), and mitochondrial amidoxime reducing component (mARC) [70], also being a reversible activator of adenylyl cyclase [71,72,73] in animals. Additionally, Mo is a co-factor of nitrate reductase in all plants and nitrogenase in Fabaceae plants associated with bacteria-fixing atmospheric nitrogen [74,75]. These enzymes possess a unique co-factor (Mo co-factor—Moco) that contains a novel ligand in bioinorganic chemistry, the pyranopterin ene-1,2-dithiolate [76]. Moco comprises Mo covalently bound to two S atoms of a unique tricyclic pterin moiety called molybdopterin [77] (Figure 1). It is still noteworthy that S required for Moco biosynthesis is obtained from Cys.

For humans, Mo is required in small amounts; however, it is worth considering that studies that assess daily nutrient requirements are performed onhealthy individualsand do not take aging into account. The tolerable upper intake level for the United States and Canada was set at 2 mg day^−1^ [31]. The European Commission suggested an upper limit of 0.6 mg day^−1^ [78]. Healthy Mo serum ranges are 0.28 µg L^−1^–1.17 µg L^−1^ [79]. Giussani et al. [80] verified that Mo is rapidly and efficiently absorbed into the systemic circulation. When it is administered dissolved in black tea and composite meals, the absorption rate is lower and delayed than dissolved in water.

The Mo deficiency results in high levels of sulfite and xanthine and low levels of SO_4_^2−^ and uric acid in the blood and urine due to impaired SUOX and XO activity, respectively [81]. XO is an enzyme that generates reactive oxygen species, such as superoxide and peroxide, by reacting with O_2_ [82]. This enzyme catalyzes the oxidation of hypoxanthine to xanthine and can further catalyze the oxidation of xanthine to uric acid.Moreover, XO plays an essential role in purine catabolism in some species, including humans, impacting adenosine and guanosine production [83,84]. Hereditary xanthinuria (type I) is caused by an inherited XO deficiency characterized by low blood and urine uric acid and high urinary xanthine contents [85].

Alonso-Andrés et al. [86] found significantly decreased adenosine, guanosine, hypoxanthine, and xanthine contents in the frontal cortex from the early stages of AD pathology. Additionally, Du et al. [87] showed using meta-analysis that those patients with AD had lower uric acid contents than healthy controls and verified that high serum uric acid contents were significantly associated with decreased risk of AD. The authors concluded an inverse association between serum uric acid contents and AD risk. This allows us to hypothesize that a lower XO activity, maybe linked to Mo deficiency, correlates to higher AD risk. Interestingly, Tana et al. [88] reviewed the related literature and concluded that serum uric acid might modulate cognitive function. Then, current studies indeed demonstrate that uric acid may exert neuroprotective actions in AD. In addition, Latourte et al. [89] commented that patients with gout (a form of inflammatory arthritis characterized by persistently elevated contents of uric acid in the blood) may have decreased risk for AD.

The reaction catalyzed by SUOX is the terminal step in the excretory metabolism of S-containing amino acids such as Met and Cys, by which sulfite produced from Met and Cys metabolism is detoxified [90]. Additionally, via the H_2_S pathways, obtained during metabolism of Hcy in Cys, significant quantities of sulfite are formed that require oxidation by SUOX [91]. In the nervous system, H_2_S acts as a neuromodulator influencing synaptic activity under regulation of steroid hormones and neurotransmitters; it may be involved in associative learning [92]. An interesting occurrence is that H_2_S in the brain is severely reduced in AD [93]. These authors commented that the reduction of H_2_S may be involved in some aspects of the cognitive decline associated with AD.

SUOX is the key enzyme in SO_4_^2−^ production. Due to low SUOX activity, biochemical abnormalities include high urinary sulfite, thiosulfate, and S-sulfocysteine excretion, which are abnormal metabolic products of degradation of S amino acids and low SO_4_^2−^ content [9,10]. SUOX is located in the mitochondrial intermembrane space [74]. Cytochrome c is the electron acceptor for the SUOX [94], with oxygen and potassium ferrocyanide as alternative oxidizing substrates. Bosetti et al. [95] studied COX of isolated mitochondria from platelets and postmortem motor cortex and hippocampus from AD patients and age-matched control subjects. Compared with controls, COX activity from AD was decreased significantly in platelets (−30%, *p* < 0.01, *n* = 20) and the hippocampus (−35 to −40%, *p* < 0.05, *n* = 6). Similarly, Maurer et al. [96] also worked with postmortem brain tissues and found a selective defect of COX in the AD brain tissue versus the normal human brain.

In general, the low SUOX activity can be due to problems in the enzyme synthesis or problems in the Moco synthesis [97]. The lowest synthesis of the enzymes or Moco occurs in newborns or infants, and these are acute health problems that can lead to death [98]. However, it alsoappears as “late-onset” or “mild” [99]. A third explanation for low SUOX activity may be a Mo deficiency.

Every unbalanced chemistry candidate contributor to AD, such as lower Mo contents, is caused by other metabolic, physiological, or dietary imbalances. Given that Mo deficiencies are very rare, there is a need for further studies prior to concluding that they are the cause of AD. The only link between Mo levels and AD that observed is that SUOX, a Mo-containing protein, could be dysregulated in AD. High levels of sulfites due to low levels of SUOX could dysregulate glutaminergic neurotransmission, contributing with one of the main aspects of AD [100].

We suggested relationships between the actions of Mo as a co-factor or enzyme stimulator and metabolites to which it is related and AD. There are many similarities between Mo deficiency and AD metabolic symptoms (Table 1).

In addition, Xia and Storm [101] reported that adenylyl cyclases, type 1 and type 8, are essential during memory formation. It is important to emphasize that AD patients/models demonstrate reductions in adenylyl cyclase activity [102,103]. Nonaka et al. [104] recently reviewed the interactions between adenylyl cyclase and dementia.

The hypothesis regarding a link between Mo and Aβ is indirect, because Mo is not directly involved in the formation of Aβ in the same way as Cu. However, Kong et al. [108] found that, similar to MoS_2_ nanoparticles, MoS_2_ nanosheets possessed therapeutic effects on AD by inhibiting Aβ aggregations and degrading the previously formed Aβ fibrils. Similarly, Manna et al. [109] found the disappearance of fibrillar structures and recovery from neurodegenerative disorders in molybdate-treated Aβ 42-mutant Drosophila flies as compared to the untreated ones, corroborating the therapeutic ability of (NH_4_)_6_Mo_7_O_24_ for treating AD. Thus, it is possible that, in normal conditions, Mo may have a destabilizing effect on Aβ fibril formation and that its deficiency may impair this natural process, facilitating Aβ formation.

## 6. Linking Metabolic Interaction of Cu, Mo, and S to AD Pathology or Multiple AD-Associated Factors

Many researchers have studied the interaction of Cu–Mo–S in non-ruminant nutrition. The majority of the studies are about dietary Mo excess causing depletion in Cu contents in rats, swine, etc. [110,111]. Conversely, studies demonstrated that excess Cu causes Mo deficiency. In rats with excess Cu supply, a non-absorbable complex of Cu and Mo in the gastrointestinal tract might reduce Mo absorption [112]. Suttle [113] commented that it is possible that at low Mo concentrations, the primary site of Cu–Mo interaction is in the gut, resulting in a decreased uptake of Cu. In contrast, at high Mo contents, high levels of Mo in the bloodstream and tissues favor the formation of inorganic and organic Cu–Mo complexes, which accumulate in the tissues.

Experimental studies indicate that Cu from the Cu–Mo complex is unavailable for ceruloplasmin synthesis. Dowdy and Matrone [114] showed a lower level of ceruloplasmin activity in the serum of pigs receiving the Cu–Mo complex (in a value very close to those not receiving Cu) than with Cu sulfate-fed controls. Dowdy et al. [115] have suggested that this Cu–Mo complex might be the mineral lindgrenite (2CuMo0_4_. Cu(OH)_2_). Arthur [116] reported lower Mo content in the liver of Guinea Pigs supplemented with Cu. They found 3.4 ± 0.4 and 6.2 ± 1.0 mg kg^−1^ of Mo in the liver of animals fed with 10 mg kg^−1^ or no Cu supplemented diet, respectively.

In vitro studies indicated a competition between high levels of Cu and Mo during Moco formation [97,117]. Inhibition of Moco synthesis by Cu could be explained by inhibiting the Mg-dependent Mo insertion reaction [97]. Hadizadeh et al. [117] found stimulation and inhibition of XO activity by Cu^2+^. Less than 5 µM Cu^2+^ resulted in stimulating the XO enzymatic activity, while in a concentration-dependent manner, 5–700 µM, 700 to 2000 µM, and 2000 µM inhibited moderate, drastic, and ultimately, respectively. Mendel and Kruse [118] commented that Moco deficiency could not be seen as an isolated defect but in a context with cellular Cu metabolism since increased cellular Cu contents might be causal for a decreased rate of Moco synthesis [97].

Suzuki et al. [119] investigated Cu–S metabolism interaction. They concluded that nCp–Cu ions in the bloodstream bind primarily to the significant form of albumin to form the mercaptalbumin-Cu complex. The latter sequesters Cys to form the albumin–Cu–Cys complex. Only Cu in the albumin–Cu–Cys complex must be transferred to the liver with albumin remaining in the blood serum as an albumin–Cys complex.

A high Cu/Mo ratio is present in children with Wilson disease. Mahoney et al. [120] demonstrated decreased uric acid production, suggesting reduced XO activity and possible Mo deficiency. The deleterious effect of high ratio Cu/Mo with impact on low XO activity has been commented on in human nutrition since many years ago [121]. Seelig [121] highlighted excess Cu levels in blood and liver associated with decreased Mo levels and/or low XO activity.

Bar-Or et al. [122] demonstrated the relationship between Cu and S metabolism has a direct effect in AD. They show that high nCp–Cu causes Hcy accumulation due to a decrease in Cystathionine b-synthase (CBS) activity: 10 µM and 25 µM Cu decreased CBS activity by 50% and 70%, respectively. Additionally, low CBS activity results in low Cys synthesis and consequently low glutathione (GSH) synthesis, since Cys is the amino acid rate-limiting in its production. GSH concentration is reduced in MCI and AD in brain regions (hippocampi and frontal cortices) affected by AD pathology [58]. Another effect is a low synthesis of H_2_S endogenously produced in the brain from Cys by CBS. H_2_S in the brain is severely reduced in AD, and, as mentioned, its reduction may be involved in cognitive decline because it functions as a neuromodulator [93].

Table 2 summarizes possible links between AD and Mo deficiency or/and Cu–Mo–S altered metabolism.

## 7. Cu and Mo Increase in the Blood Vessels and Evidence Sustaining a Decreased Cu and Mo Transport in the Brain with Aging

Paglia et al. [124] evaluated four groups of 118 people aged between 54 and 87 years old (HS, healthy subjects—*n* = 34; SMC, subjective memory complaint—*n* = 20; MCI, mild cognitive impairment—*n* = 24; AD, Alzheimer disease—*n* = 40). Mo content increased in serum progressively, passing from HS through SMC, MCI, up to AD, and the difference between HS and AD was statistically significant (0.83 ± 0.26; 0.99 ± 0.24; 1.09 ± 0.36; and 1.20 ± 0.52 μg L^−1^ of Mo in HS, SMC, MCI, and AD, respectively). Additionally, serum Cu concentration was higher in SMC than in HS (*p* > 0.05) 858.96 ± 224.19 and 703.88 ± 244.03 μg L^−1^, respectively.

The results of Mo content [124] indicate that there is a progressive Mo accumulation in serum throughout the worsening phases of AD symptoms. On the other hand, Cu increase in serum is likely associated with the increase of the nCp–Cu component [43], and nCp–Cu increases the susceptibility to AD [125,126]. Additionally, Cu decreases in the brain in AD [36] and, possibly, Mo content decreases in tissues in AD, as the decreased levels of molybdoenzymes products suggest [7,87].

There are some similarities between Cu and Mo metabolism in AD, so their serum and tissues contents have a similar pattern, as shown in Table 3.

Based on the evidence they reported, they suggested a higher Mo level in serum [124] and a lower one in urine [30] and brain. Our hypothesis posits that there is little Cu–Mo complex formation inside blood vessels when there is an adequate nCp–Cu /Mo ratio in youth. Consequently, Cu and Mo are efficiently transported out of the vessels, reaching the tissue cells as depicted by Smith and Wright [127] for sheep with Cu–Mo balanced diet. With aging, people with a genetic predisposition for AD may have progressive nCp–Cu accumulation in the body that can likely increase the formation of Cu–Mo complexes inside the blood vessels. Under these conditions, the transport of Mo out of the vessels might be reduced, causing a mild Mo deficiency within the tissues. We posit that at older ages, in AD, the accumulation of nCp–Cu is increased and, consequently, the formation of the Cu–Mo complex is significantly increased inside the vessels, leading to an increase in the serum Mo content and a great decrease in the Mo transport out of the vessels, a mechanism observed by Smith and Wright [127]. This leads to an increase in the Mo deficiency symptoms, with a significant drop in the activity of enzymes that contain Mo as a co-factor, as reported by Alonso-Andrés et al. [86], Du et al. [87], Latourte et al. [89], Yamamoto et al. [102], and Kelly [103].

We propose that AD-susceptible people may have a higher nCp–Cu/Mo ratio in serum resulting in a lesser Cu/Mo ratio in food in adult age. In this case, Mo should possibly be supplemented in the daily diet. We propose future research reformulation in the daily human requirements for Cu and Mo, including the Cu/Mo ratio for different ages. For example, for ruminants, the ideal Cu/Mo ratio is >6:1 and <10:1 in the finished feed [128]. Thus, in this group of animals, if the Cu/Mo ratio is <6:1 or >10:1 in feed, it causes metabolic problems. Moreover, our discussion focuses on nCp–Cu/Mo ratio in AD blood (serum or plasma): in AD, serum nCp–Cu is higher than in healthy people [15]. The nCp–Cu/Mo ratio may be higher in serum than in healthy subjects in AD. Thus, Mo might be progressively captured by nCp–Cu and accumulated in serum, decreasing the Mo bioavailability for the Mo enzymes/proteins.

## 8. Mo Deficiency and Low SUOX and Cytochrome c Activities Are Possibly Linked to AD

As previously mentioned, in individualswith a predisposition towards AD the content of nCp–Cu in serum increases due to aging [36], which possibly causes greater Mo demands or its deficiency. One strong indication for this is the low content of molybdoenzyme products in AD [7,56,86,87,88,89]. In addition, there is an interaction of deficient Mo with disturbed S metabolism in AD [57,105].

Under Mo deficiency, the SUOX activity decreases, which leads to an increase in sulfite contents (which is toxic) [74] and endogenous S metabolism disturbance [7,54,106] with decreased contents of intracellular Cys [57] and decreases contents of SO_4_^−2^, folate, choline, and H_2_S.

The terminal step in the excretory metabolism of Met and Cys needs SUOX. If SOUX does not work well, toxic sulfite increases, and the pathway for Cys synthesis decreases. [105]. Accordingly, lower intracellular Cys confirm this idea in AD brain [57]. As Cys is required in CoA [129] and choline synthesis, the acetylcholine production is also lower, which is consistent with disturbed cholinergic neurotransmission seen in AD. Another disturbance is the excess of Met, which can increase Aβ neurotoxicity [52]. High Hcy contents in AD result from the low activity of the trans-sulphuration pathway [130]. When in low SUOX activity, sulfite loading blocks the standard conversion of Met to Cys, and this causes Cys values to fall, possibly due to accumulated sulfite [105]. We commented that Cys contents are high in the plasma in AD [7], but it may be decreased inside the brain cells [57].

In our view, the possible primary link between S and Mo metabolic interaction in AD is the low SUOX activity. It is well documented that low SUOX activity causes neurological damage [74]. Anything that negatively affects the activity of SUOX will also affect the activity of Cytochrome c. That is because sulfite induces Cytochrome c release from brain mitochondria in the presence of Ca^2+^ [131]. In addition, given that Cytochrome c and SUOX are linked, it makes sense to think that what negatively affects Cytochrome c activity will also negatively affect SUOX activity. Thus, the SUOX–Cytochrome c complex has to be working well for neurological functions to occur satisfactorily. In addition, in AD, there is a decrease in SO_4_^2−^ content [7], indicating low SUOX activity due to possible Mo deficiency.

We must consider that newborns’ acute low SUOX activity results in low plasma Hcy contents [132]. On the other hand, increased plasma Hcy contents are a risk factor for AD, and a typical AD symptom is high plasma Hcy contents [54,65,68]. The high Hcy contents in AD may be due to the direct high nCp–Cu effect demonstrated by Bar-Or et al. [124] when they showed that high nCp–Cu causes Hcy accumulation due to a decrease in CBS activity.

In conclusion, we hypothesize that the excessive nCp–Cu in the body can result in a higher requirement for Mo, which becomes deficient. It results in low SUOX activity, disrupted Met metabolism and higher Aβ content. Consequently, the excess of Met can increase Aβ neurotoxicity [52]. So, although Mo is not directly involved in the formation of Aβ, such as Cu, the link with the Aβ hypothesis is indirect (Figure 2). Its harmful effects on metabolism trigger the other symptoms of AD-related diseases and, later, AD symptoms. That hypothesis is in line with the findings of the current review.

## 9. Clinical Perspective for AD and the “Cu–Mo–S Circuitry”

Despite considerable social and economic interest in AD, which make this disease one of the most studied in neuroscience, developing an efficient therapeutic protocol is very difficult. Basic research has improved the knowledge about physiopathology and metabolic pathways involved in AD. However, this knowledge was still not able to reach an effective therapy. It has been proposed as a possible strategy for a therapeutic approach to developing disease-modifying therapies (DMT), which aim to promote a significant change in the course of AD, preventing, delaying its onset, or slowing its progression [133]. However, Liu et al. [134] confirmed the effectiveness of eye-tracking in diagnosing a cognitive disorder and concluded that it is a viable marker for recognizing that.

Up to 2021, there were very few drugs approved for AD, and all of them act only in ameliorating the symptoms. None was developed to treat the causes of AD. This is mainly because the real cause for this disease is far from being ultimately discovered. The high failure rate in clinical trials clearly demonstrates this. From 2002 to 2012, the failure rate was 99.6% [133,135].

However, the FDA approved a new treatment in 2021. Aducanumab is an amyloid-targeting monoclonal antibody applied by intravenous infusions and acts on the extracellular accumulation of Aβ protein [136]. Although it might be considered the first approved disease-modifying drug for AD, its approval is highly controversial. Several researchers and scientific societies in the fieldhave stated that there is little evidence of its absolute safety and efficacy and that it has a high financial cost [137,138,139]. Therefore, despite the effort to find an effective treatment for AD, this has not been accomplishedto date.

Regarding the main point of this review, few clinical trials involving the “Cu–Mo–S circuitry” were performed. Cu chelating agents were tested, but no satisfactory results were obtained [140].

Given that the search for a pharmacologic treatment has been ineffective, new proposals have been made. Dietary supplementation has been recently proposed to treat AD, specifically those related to the “Cu–Mo–S circuitry”, exploring the known metabolic pathways involved in AD [100]. This proposal agrees with our hypotheses of the connection between Mo and AD. It might be expected that a multifactorial disease should need a multifactorial treatment, including drugs and dietary supplementation.

Another critical point to be considered is the fact that there is deep overlap in the molecular mechanisms involved in all neurodegenerative diseases. For example, the same “Cu–Mo–S circuitry” described here for AD is also engaged in PD [141,142,143]. The therapy with Cu chelating agents was also performed for PD, with no positive result [144]. Therefore, although neurodegenerative diseases have different causes and physiopathological features, their courses involve several common metabolic pathways, which might mean that the therapeutic intervention focused on the “Cu–Mo–S circuitry” would not act in the onset of the development of all neurodegenerative diseases. By the same reasoning, this circuitry might not be the origin but a converging circuitry during neurodegeneration. Future research in this field is needed to shed light on these questions.

Another point concernscholine, folate, and Mo supplementation effects in AD. Choline supplementation significantly reduced Aβ load and improved spatial memory in APP/PS1 mice [67]. Regardingfolate, the conclusions are inconsistent while supplementing the elderly with different folic acid states [145]. To date, no research related to Mo supplementation has been conducted in humans and this would bean interesting subject for future research.

## Figures and Tables

**Figure 1 ijms-23-07935-f001:**
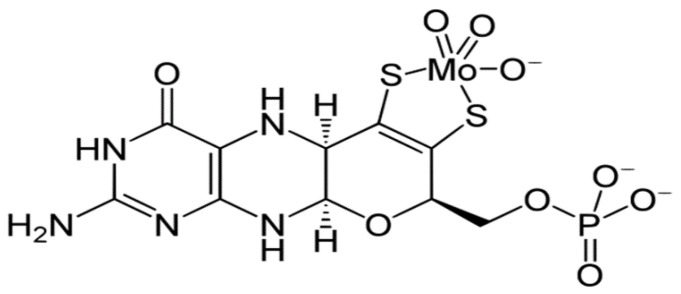
Mo co-factor—Moco.

**Figure 2 ijms-23-07935-f002:**
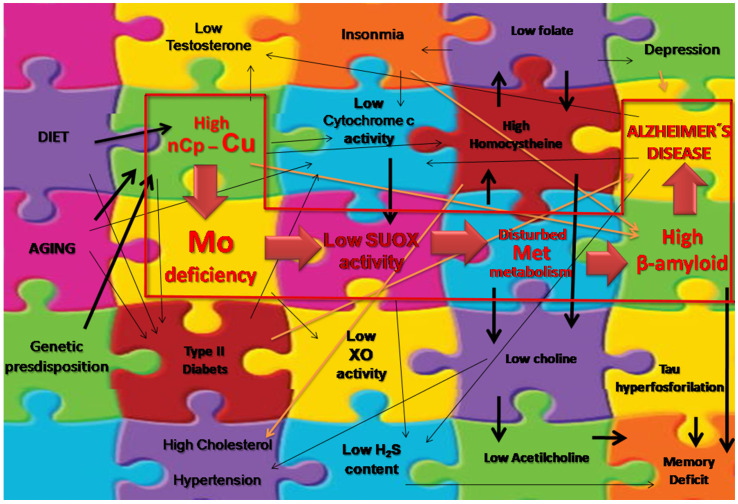
Summary of proposed cellular/molecular mechanisms for the Cu–Mo–S circuitry in AD pathology or multiple AD-associated factors.

**Table 1 ijms-23-07935-t001:** Similarities among lower activity or synthesis of XO, AO, SUOX, and Moco or Mo deficiency and symptoms of Alzheimer’s disease.

	Lower Activity or Synthesis of XO, AO, SUOX or Moco, or Mo Deficiency	Alzheimer’s Disease
	Similarities	
Sulfate	Low urinary SO_4_^2−^ content [9,10].	High Cys/SO_4_^2−^ in blood plasma [7]; AD’s plasma had only 22% SO_4_^2−^ than standard control [7].
Methionine	High plasma Met [105].	Alterations in pathways related to Met metabolism [106]; Serum levels of Met-SO are higher in early-stage AD as compared with MCI patients [53].
Cysteine	Sulfite loading stops the conversion of Met to Cys, and this causes Cys levels to fall [105].	Cys content decreased inside brain cells [60]; Cys is high in brain AD [58].
Uric acid	Serum and urinary uric acid contents are reduced [107].	Lower contents of uric acid in AD patients than healthy controls [87]; Patients with gout with elevated uric acid contents in the blood may have decreased risk for AD [89].

**Table 2 ijms-23-07935-t002:** Possible links between Alzheimer’s Disease (AD) and Mo deficiency or/and Cu–Mo–S altered metabolism.

Event	Research Results
NCp–Cu accumulation in serum or plasma.	Chronic exposure to Cu and its dyshomeostasis has been linked to accelerated cognitive decline and potentially increased the risk of AD [41,42,43]; High nCp–Cu in the serum characterizes AD patients [36]; Serum Cu was inversely associated with testosterone [23] and Rosario et al. [22] found that brain levels of testosterone are significantly lower in AD.
Cu and Mo are antagonists, so excessive Cu cause Mo deficiency and vice versa.	The formation of a Cu–Mo complex can be the place of nutritional interaction between the two metal ions [116]; Cu–Mo complex was unavailable for ceruloplasmin synthesis [117]; Excess in Cu supply and a non-absorbable complex of Cu and Mo in the gastrointestinal tract might reduce Mo absorption [114]; Reducing uptake or retention of Mo in Cu-deficient induces Cu-excess in rats [114]; Increased cellular Cu contents might be causal for a decreased rate of Moco synthesis [97]; Competition between Cu, at a high level, and Mo occurs during Moco formation [119]; Cu/Mo ratio imbalance affects S metabolism [123].
Products of enzymes’ activity that have Mo as a co-factor are in lower content in AD than in healthy people	A high Cu/Mo ratio affects xanthine oxidase activity [121]; Decreased contents of adenosine, guanosine, hypoxanthine, and xanthine in early AD frontal cortex [86]; High serum uric acid contents were significantly associated with decreased risk of AD, so patients with gout may have reduced risk for AD [86,89]; AD patients/models demonstrate reductions in adenylyl cyclase activity [102,103].
Altered S amino acids metabolism occurs in AD	Mo is the sulfite oxidase (SUOX) co-factor: its low activity damages S metabolism [9,10]; AD has a high Cys/SO_4_^2−^ ratio in plasma [7].; In AD, there is a plasma decrease of SO_4_^2−^ level [7] (low SO_4_^2−^ content indicates low SUOX activity due to possible Mo or Moco deficiency); Cys levels are high in AD plasma [7]; Cys decreased by ca. 3-fold in the brain of a rat model of AD [57]; Cytochrome c is the electron acceptor for the SUOX. COX activity from AD was decreased significantly in platelets and the hippocampus [95]; High nCp–Cu causes an accumulation of Hcy due to a decrease in Cystathionine b-synthase activity [122].
Mo contents	AD: serum Mo level increased progressively, passing from healthy subjects (HS) through subjective memory complaint, mild cognitive impairment up to AD [124]; Diabetic (T2D) with severe complications: serum Mo contents increased [30]; T2D with severe complications: lower urine Mo contents [30].

**Table 3 ijms-23-07935-t003:** Similarities between Cu and Mo contents in Alzheimer’s disease (AD).

	Serum Levels	Brain Tissue Levels
	Similarities	
Cu	High content of nCp–Cu in serum [36].	In brain tissues, the Cu content is usually low [36,68].
		Evidence suggesting a low level of Mo in AD brain tissues:
Mo	Serum Mo content increased progressively, passing from healthy subjects (HS) through subjective memory complaint, mild cognitive impairment up to AD, and the difference between HS and AD was statistically significant [124].	High serum uric acid contents were significantly associated with decreased risk of AD, so patients with gout may have decreased risk for AD [87,89]; Brain tissues from AD patients/models demonstrate reductions in adenylyl cyclase activity [102,103].

## Data Availability

Not applicable.

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
