# Peer review of "Is There a Connection between the Metabolism of Copper, Sulfur, and Molybdenum in Alzheimer’s Disease? New Insights on Disease Etiology"

_ijms, 2022, doi:10.3390/ijms23147935_

Round 1

Reviewer 1 Report

In their manuscript, authors hypothesize that the excessive nonceruloplasmin-bound Cu in the body can lead to the formation of Cu-Mo complex in blood vessels, which cannot be transported out from the vessels and results in a higher requirement for Mo for tissues, which becomes deficient. Although the intracellular deficiency of Mo it is not shown, it is further hypothesized that it results in the low activity of Mo-dependent sulfite oxidase and low SO42- and toxic sulfite content, which triggers other AD symptoms.

The proposed hypothesis is fascinating and aims to identify molecular processes leading to Alzheimer’s disease, however, by raising hypotheses we should proceed with caution, in order to avoid speculation. In the biometal field, even statistically significant associations do not provide direct evidence of a causal relationship between metal levels and Alzheimer’s disease, because a dyshomeostasis of metals could be behind the pathogenesis of this disease also without being its main causal factor.

Besides that, the main shortcoming of the hypothesis is connected with the fact that no relations with the amyloid cascade is provided. Today it is proven that amyloid formation is the central molecular pathway of AD, which leads to the formation of neurofibrillary tangles and neurodegeneration. We think that each new hypothesis about the reasons of AD should currently consider links of putative factors with amyloid cascade. We suggest to take an example from Ref. 8 (Squitti, R.; Faller, P.; Hureau, C.; Granzotto, A.; White, A. R.; Kepp, K. P. Copper Imbalance in Alzheimer's Disease and Its Link with the Amyloid Hypothesis: Towards a Combined Clinical, Chemical, and Genetic Etiology. J Alzheimers Dis. 485 2021, 83(1), 23-41), where such a link in the case of copper is considered. Moreover, this reference is an example of how hypotheses about the relation between metal metabolism and AD has to be elaborated.  Therefore, focusing further studies on a mechanism of action of molybdenum, but also other less studied metals like for example chromium and manganese, would be needed for raising hypotheses about their role in AD.

Other points:

·        Manuscript is not review but rather hypothesis.

·        There are too many weakly linked data included.

·        Figure 2 is too simplified.

We suggest reorganizing the manuscript to a shorter version and publishing it as the hypothesis, where the link with the amyloid cascade is also considered.

Author Response

Reviewer 1

.  

  1. Manuscript is not review but rather hypothesis.

      Our review shows complex relationships between copper (Cu), molybdenum (Mo) and sulfur (S) metabolism and AD pathology or multiple AD-associated factors. Our new hypothesis is scientifically based on this review. We think that the structure presented might allow a greater understanding of the complex metabolic phenomenon. So, we organized our manuscript mentioning each part of these metals metabolism in which its perturbation could lead potentially to AD pathology or AD-associated factors in order to provide subsidies to raises the hypothesis.

  1. There are too many weakly linked data included.

We have initially presented the relationships of Cu, Mo and S and in subsequent items they presented links of Cu, Mo and S metabolism related to AD pathology. Figure 3, added at the request of reviewer 2, summarizes these complex relationships and highlights the Cu-Mo-S circuitry. The number of data included is high because is a complex study about metabolic links of Cu, Mo and S related to AD pathology or multiple AD-associated factors.

  1. Figure 2 is too simplified.

We improved figure 2 and added a caption to the figure title for clarity.

  1. We suggest reorganizing the manuscript to a shorter version and publishing it as the hypothesis, where the link with the amyloid cascade is also considered. (... an example from Ref. 8 (Squitti, R.; Faller, P.; Hureau, C.; Granzotto, A.; White, A. R.; Kepp, K. P. Copper Imbalance in Alzheimer's Disease and Its Link with the Amyloid Hypothesis: Towards a Combined Clinical, Chemical, and Genetic Etiology. J Alzheimers Dis. 485 2021, 83(1), 23-41), where such a link in the case of copper is considered. Moreover, this reference is an example of how hypotheses about the relation between metal metabolism and AD has to be elaborated)

Thanks for the suggestion. We read with great interest the ref 8. However, this Journal does not support a specific format for ‘hypothesis’ study. Our new hypothesis is scientifically based on a review of the complex relationships between copper (Cu), molybdenum (Mo) and sulfur (S) metabolism and AD pathology or multiple AD-associated factors. We think that the structure presented might allow a greater understanding of the complex metabolic phenomenon.

Regarding the A-beta connection, we could roughly summarize as follows: AD patients have higher Cys and Met-SO serum levels, and a higher content of sulfonated amino acids is noticed in Aβ-plaques. Mo, the active site in Sulfite oxidase (SUOX), is deficient in AD patients. So, although Mo is not directly involved in the formation of Aβ-plaques such as Cu, the link with the amyloid hypothesis is indirect. This indirect link has been explained and improved (in red) in the manuscript in sections 4, 5 and 8. 

We improved that discussion on page 6 and 7: “The hypothesis regarding a link between Mo and Aβ is indirect, because Mo is not directly involved in the formation of Aβ in the same way as Cu. However, Kong et al. [108] found that, similar to MoS2 nanoparticles, MoS2 nanosheets possessed therapeutic effects on AD by inhibiting Aβ aggregations and degrading the previously formed Aβ fibrils. Similarly, Manna et al. [109] found the disappearance of fibrillar structures and recovery from neurodegenerative disorders in molybdate-treated Aβ 42-mutant Drosophila flies as compared to the untreated ones, corroborating the therapeutic ability of (NH4)6Mo7O24 for treating AD. Thus it is possible that, in normal conditions, Mo may have a destabilizing effect on Aβ fibril formation and that its deficiency may impair this natural process, facilitating Aβ formation”.

Reviewer 2 Report

The manuscript by Coelho et al. reviewed the literature for the past 60 years on plausible links between Cu, S, and Mo elements in AD. Because AD and related dementias are attributed to complex and multifaceted pathophysiological mechanisms, this review paper provides a new perspective in the field of AD research and is worthy of dissemination.

Minor Comments:

1)     Abbreviations (e.g., AD, PD, Cu, S, Mo) should be clearly defined when they are first mentioned in the text. A section of Abbreviations should be appended in the end of the manuscript.

2)     The authors emphasized on an imbalance of Cu, Mo and S elements in disease conditions. It would be also good to know whether nutritional/therapeutic management of these elements can help with AD. The author should comment on that.

3)     Showing a summary figure of proposed cellular/molecular mechanisms for the Cu-Mo-S circuitry in AD or related neurodegenerative diseases in general would be better, which could help readers to capture key ideas of the paper.

4)     Regarding English language and style, in general the manuscript is fine and easy to follow, but there are a few typos/poor grammars need to be corrected throughout the main text.

Author Response

Reviewer 2

  1. Abbreviations (e.g., AD, PD, Cu, S, Mo) should be clearly defined when they are first mentioned in the text. A section of Abbreviations should be appended in the end of the manuscript.

We reviewed all abbreviations in the manuscript to ensure that they are defined when first mentioned and we included a list of abbreviations in the end of the manuscript, as requested  (see page 13).

  1. The authors emphasized on an imbalance of Cu, Mo and S elements in disease conditions. It would be also good to know whether nutritional/therapeutic management of these elements can help with AD. The author should comment on that. “Another point is choline, folate, and Mo supplementation effects in AD. Choline supplementation significantly reduced amyloidβ plaque load and improved spatial memory in APP/PS1 mice [143]. About folate, the conclusions are inconsistent, while supplementing elders with different folic acid states [144]. To date, no research related to Mo supplementation has been conducted in humans and this is an interesting subject for future research.”

This was done as requested (see page, 13).

  1. Showing a summary figure of proposed cellular/molecular mechanisms for the Cu-Mo-S circuitry in AD or related neurodegenerative diseases in general would be better, which could help readers to capture key ideas of the paper.

This was done as requested. We added Figure 3 that summarizes the complex relationships and highlights the Cu-Mo-S circuitry.

  1. Regarding English language and style, in general the manuscript is fine and easy to follow, but there are a few typos/poor grammars need to be corrected throughout the main text.

This was done as requested. The manuscript was reviewed by a native English speaker.
